# Neuroprotective Role of Lactoferrin during Early Brain Development and Injury through Lifespan

**DOI:** 10.3390/nu14142923

**Published:** 2022-07-17

**Authors:** Gabriel Henrique Schirmbeck, Stéphane Sizonenko, Eduardo Farias Sanches

**Affiliations:** 1Biochemistry Post-Graduate Program, Biochemistry Department, Federal University of Rio Grande do Sul, Porto Alegre 90035-003, Brazil; gschirmbeck@gmail.com; 2Division of Child Development and Growth, Department of Pediatrics, School of Medicine, University of Geneva, 1205 Geneva, Switzerland; ef.sanches@yahoo.com

**Keywords:** brain development, prematurity, lactoferrin, neuroprotection, neurodegenerative diseases

## Abstract

Early adverse fetal environments can significantly disturb central nervous system (CNS) development and subsequently alter brain maturation. Nutritional status is a major variable to be considered during development and increasing evidence links neonate and preterm infant impaired brain growth with neurological and psychiatric diseases in adulthood. Breastfeeding is one of the main components required for healthy newborn development due to the many “constitutive” elements breastmilk contains. Maternal intake of specific nutrients during lactation may alter milk composition, thus affecting newborn nutrition and, potentially, brain development. Lactoferrin (Lf) is a major protein present in colostrum and the main protein in human milk, which plays an important role in the benefits of breastfeeding during postnatal development. It has been demonstrated that Lf has antimicrobial, as well as anti-inflammatory properties, and is potentially able to reduce the incidence of sepsis and necrotizing enterocolitis (NEC), which are particularly frequent in premature births. The anti-inflammatory effects of Lf can reduce birth-related pathologies by decreasing the release of pro-inflammatory factors and inhibiting premature cervix maturation (also related to commensal microbiome abnormalities) that could contribute to disrupting brain development. Pre-clinical evidence shows that Lf protects the developing brain from neuronal injury, enhances brain connectivity and neurotrophin production, and decreases inflammation in models of perinatal inflammatory challenge, intrauterine growth restriction (IUGR) and neonatal hypoxia-ischemia (HI). In this context, Lf can provide nutritional support for brain development and cognition and prevent the origin of neuropsychiatric diseases later in life. In this narrative review, we consider the role of certain nutrients during neurodevelopment linking to the latest research on lactoferrin with respect to neonatology. We also discuss new evidence indicating that early neuroprotective pathways modulated by Lf could prevent neurodegeneration through anti-inflammatory and immunomodulatory processes.

## 1. Fueling the Brain during Early Neurodevelopmental Stages

Human brain energetic demand is always a subject of interest and inspires awe: despite weighing around 2% of an adult human body, it requires 15–25% of total glucose consumption in a resting condition, which is a striking difference when compared to other vertebrates, whose CNSs demand 2–10% of energy requirements [1,2]. Although allocating almost a quarter of total glucose demand to a single organ is a remarkable endeavor, this accounts only for resting state measurements in adults, where the energy is required for housekeeping functions, such as maintenance of ionic gradients, neuronal repolarization, neurotransmitter uptake and recycling, membrane and organelle maintenance and protein turnover [3,4,5]. In newborns and young children, these values are still unclear [6]; however, they are possibly much higher in comparison to adults. Besides greater relative brain mass [7], the newborn brain undergoes a great increase in both volume (from 200 cm³ to 600 cm³) and surface area (from 700 cm² to 2000 cm²) due to gyrification and architectural changes during the first two years of life [8]. These modifications require high energy and macronutrient quantities that we are still unable to determine precisely, since, apart from the estimated brain biomass accretion, other vital processes also take place that demand energy. These include architectural changes, cortical layering, cellular differentiation, synaptic plasticity and maturation requiring genomic regulation and proteome turnover, which present technologies are unable to assess in in vivo systems.

Neurodevelopment is substantially affected by environmental conditions [9], and nutrition is arguably the most important environmental need during development. Adequate nutrition during pregnancy and infancy lay the foundations for neuroplasticity and ultimately the development of cognitive, motor, and socio-emotional skills throughout childhood and adulthood. In the first thousand days after conception, the exponential growth experienced by the brain requires prioritizing nutrients that can maximize neural health and development, which is crucial for nutrient interventions that can impact lifelong brain function [10,11,12,13,14]. Thus, investigating nutritional needs, and proposing new nutritional strategies for the period of brain development, recognizing that fetal and early-life programming could exert a long-term influence on brain maturation, has the potential to be a cost-effective approach that may be encouraged by public health policies [15,16]. 

## 2. Nutritional Needs during Neurodevelopment

Essential fatty acids, iron, zinc, iodine, vitamins B12 and D are particularly important for the formation of the neural plate and neural tube up to early childhood [17]. Cell division begins within the neural tube, providing the origin for neurons and glial cells. Later in gestation, interaction with scaffolding cells, such as astrocytes, microglia, radial glia, and oligodendrocytes, leads to the initiation of neuronal migration (the peak of which is reached from the third to the fifth month of gestation). The establishment of neural networks, which enable neural processes (dendritic and axonal ramifications) to communicate, are now formed and functional synaptic transmission can be initiated (the synaptogenesis peak is reached at around 34 weeks of gestation) [18]. In the last trimester of gestation, myelination starts and continues through adolescence and into adulthood [19]. Due to the intricate nature of these processes, specific nutritional inadequacy during critical stages of CNS can impair the proper development of sensory, motor, and cognitive skills [20,21]. However, nutrients are vital to the brain, not only for morphological growth, but also for ongoing neurochemical and neurophysiological processes [20]. The brain is a heterogeneous organ comprised of completely distinct anatomical regions (such as the hippocampus and cortex), processes (e.g., myelination, neurotransmitter and neurotrophic factor production, cell migration and synaptic pruning) and metabolic demands that vary according to the region, and the time of development, and have specific nutrient requirements. Moreover, nutrition has emerged as a potential factor, not only to improve growth and outcomes by optimizing early nutritional intake, but also as a determinant of improved neurodevelopment through enhanced brain maturation, and has been shown to prevent future neuropsychiatric disorders [22,23,24].

Inadequate nutrition can lead to detrimental effects on maternal physiology and health (e.g., placental insufficiency, preeclampsia, diabetes, anemia), which, in turn, can affect the offspring’s development (e.g., stillbirth, growth delay, risk of developing disorders, fetal growth restriction). In general, the higher the rate of an organ’s growth, the greater its vulnerability to insufficient nutrient supply. This makes the developing brain highly susceptible to injury with an accretion rate of 7 g/day of fat and 2 g/day of protein between 25 and 34 gestational weeks. Deficiency or excess of macronutrients, such as proteins, are most important during the second and third trimesters of gestation, when growth and development of fetal tissues is accelerated [15]. Their deficiency affects several body systems and can cause impaired brain growth and long-term neurocognitive deficits [24,25,26]. Protein-energy malnutrition in human fetuses between 24 and 44 weeks of gestational age can result in intrauterine growth retardation and is usually linked to maternal hypertension or severe malnutrition. Interestingly, protein-energy malnutrition in fetal and early neonatal animal models has been shown to reduce neural deoxyribonucleic (DNA) and ribonucleic (RNA) acid content, resulting in smaller brains, fewer neurons, and reduced synapse number and dendritic arbor complexity due to reduced protein synthesis and hypomyelination [27,28,29]. This is particularly relevant in the context of premature delivery, since evidence has shown that the composition of maternal milk differs in mothers who deliver at term vs. preterm. Preterm newborns receive milk having increased levels of proteins, fat, free amino acids, and sodium, but, over the first few weeks following delivery, these levels decrease [30,31]. Carbohydrates are involved in energy supply, glycemia control, insulin metabolism, cholesterol and triglyceride metabolism [29,30] and are the main energetic source sustaining pregnancy and lactating processes [30,31]. Carbohydrate and fat overconsumption can induce CNS pro-inflammatory profiles [30,32]. Evidence shows that carbohydrates, especially dietary fiber, contribute to the maintenance of gut microbiome diversity [33,34,35]. High fat level diets can induce maternal overweight or obesity and diabetes and impaired myelination through a sustained proinflammatory state [15,36,37,38]. Polyunsaturated fatty acids (PUFAs) are an essential type of fat, especially required during pregnancy and breastfeeding. α-linolenic acid (n-6 PUFA) and linoleic acid (n-3 PUFA) requirements are high in pregnancy, notably from the third trimester of gestation up to the first two years after birth, since PUFAs have been shown to contribute to brain growth [39,40], neurogenesis increase (*in vitro* and *in vivo*) [41,42], and increase in synaptic plasticity and neuronal wiring [43,44]. Dietary fats can also be used to produce ketone bodies as a supplemental metabolic substrate instead of glucose to supply the developing brain [45]. 

Among essential micronutrients, those with anti-inflammatory properties are the most commonly present. Vitamin A and its active oxidized forms, retinaldehyde and retinoic acid, are involved in development, growth, immunity and during pregnancy, where they are needed by the mother for placental maintenance and by the embryo for formation and development of the spinal cord and brain [46,47,48,49]. Vitamin B9 or folate (or its synthetic form, folic acid) is involved in the synthesis of DNA, RNA, and some amino acids [26,50,51], and methylation, and is important during periods of placentation, implantation of the embryo, embryogenesis, and fetal growth [52,53]. Furthermore, folate is very well known in the prevention of neural tube defects—interestingly, low maternal folate status has been associated with preterm birth and growing evidence links maternal folate deficiency to autism spectrum disorders and schizophrenia [54,55]. Vitamin B12 acts in DNA methylation [52,53] and is also involved in lipid metabolism [56]. B12 deficiency has been associated with higher levels of neuroinflammation and oxidative stress [55]. Vitamin D and calcium are closely related in terms of metabolism and are critical for immune and inflammatory functions, as well as cellular differentiation [56]. Vitamin D has been shown to attenuate lipopolysaccharide (LPS)-induced inflammation and can also modulate the proliferation, differentiation, survival, maturation, and cytokine release of several immune cells [57]. Excessive sodium chloride consumption during pregnancy can cause CNS disorders and inflammation [58,59]; however, studies of the maternal-fetal effects of a high-salt diet on neurodevelopment are scarce. Diets with a high concentration of salt have a proinflammatory profile, exacerbating immune challenges induced by LPS [60] and exacerbating the onset of immune diseases (e.g., colitis, *lupus erythematosus*, lung injury) [60,61,62]. Iodine is a common mineral deficiency observed in the diet of pregnant women [62], despite its antioxidant properties [63]. Iodine is critical for the fetal and newborn modulation of cellular migration and differentiation, synaptogenesis and myelination around the second trimester of pregnancy, and later in life due to its role in thyroid hormone synthesis [64,65]. Iron deficiency is the most common nutrient alteration during pregnancy. Iron is essential for heme protein synthesis and oxygen transport [66] and has been shown to modulate inflammatory processes through the reduction of anti-inflammatory cytokines and mediators, as well as by inducing the upregulation of pro-inflammatory cytokines [67]. Excessive levels of iron are detrimental for cardiovascular system health leading to endothelial oxidative stress, which is particularly harmful for the developing brain. Zinc is necessary for cellular processes during division, differentiation, and function and is thus crucial to embryogenesis, fetal growth, and development, as well as milk production [68,69]. Zinc is also an essential mineral for intestinal microbiome flora health [70]; its deficiency during pregnancy induces neuroinflammation, inhibits signaling molecules, such as nuclear factor kappa B (NF-kB), and prevents cytokine production. Zinc closely regulates metalloproteinases 2 (MMP-2) and 9 (MMP-9) and PPAR-α [71,72]. Low zinc levels increase the expression of interleukin 6 (IL-6) and astrogliosis in the offspring’s brain [70,72], with attention-deficit hyperactivity disorder (ADHD), hypotonia, seizures [73] and the development of autistic-like behavior being reported in preclinical models [70]. 

Thus, specific nutritional deficits trigger cellular and molecular cascades that can induce neural degeneration and block or delay maturation of oligodendrocyte progenitor cells leading to demyelination and, later, to impaired connectivity that can lead to permanent motor, cognitive, and/or behavioral deficits. Here, we have shown that the nature and severity of changes vary according to the type and severity of nutritional insult and the maturation stage of the brain. Based on these observations, strategies focused in nutritional modulation can impact the functioning of glial cells (in particular microglia) [74] and attenuate brain injury (in particular, neuroinflammation) and are key for perinatal brain protection as summarized in Figure 1.

## 3. Lactoferrin for Preventing IUGR/Premature Delivery and Associated Brain Injury

CNS damage and poor neurodevelopmental outcomes due to preterm delivery are worsened by the pro-inflammatory profile observed in pregnancies and babies [75]. Maternal infection, placental insufficiency leading to intrauterine growth restriction (IUGR) and sepsis, and necrotizing enterocolitis (NEC) are commonly observed in preterm deliveries and participate in the inflammatory challenge fetus and preterm newborns are subjected to [76]. Preterm babies are also prone to brain damage due to the disbalance between production and scavenging oxidative species since their antioxidant systems are not fully developed and their brains have high levels of free iron [77,78]. In this context, dietary interventions based on antioxidant and anti-inflammatory properties could represent a valuable therapeutic tool for preterm infants, not only for their direct effects in reducing preterm delivery itself, but also due to their potential for decreasing future direct brain damage and comorbidities. Lactoferrin (Lf), a physiological compound produced by exocrine glands is released at high levels in colostrum and maternal milk [76,79] and performs numerous biological functions, including as an iron chelator, an anti-inflammatory agent, an immunomodulator, and an antioxidant and plays a major role in host-defense mechanisms [80,81,82] with high concentrations found in human milk [79,82]. Due to its iron-binding properties, lactoferrin can exist in the so-called *Apo* (iron-free) state or the *holo* state, when saturated with iron, with a mixture of both being observed in milk [83,84]. This distinction is important because apo-Lf can easily chelate iron preventing bacterial growth, whereas holo-Lf is superior in correcting iron deficiency [83]. Thus, due to its role in several pathways involved in neurodevelopment in preterm infants, lactoferrin represents an outstanding candidate for study as, to date, there is no proven strategy for protecting infants from brain injury and dysfunction. 

Despite preclinical evidence showing that Lf can prevent damage due to perinatal infections, only a small number of studies have reported anti-inflammatory effects in pregnant animals. Infection/inflammation are well-known risks for preterm birth, and it is conceivable that Lf treatment might be able to prevent it [85,86,87]. Rabbits subject to intrauterine infection with *E.coli* and treated with recombinant human Lf showed increased fetus survival rate and extended length of pregnancy compared to non-treated infected animals [88]. Non-treated animals showed inflammatory exudates, endometrial necrosis and increased levels of interleukins that were not present in treated animals. Additionally, Lf has been tested for its anti-viral properties, improving the outcomes in infection models of HSV, influenza, rotavirus and RSV. Lf appears to inhibit viral replication in vitro for HSV-1 and to reduce infection by the SARS-CoV 1 and 2 viruses, in both cases by interacting with cell surface proteoglycans, and could serve as a potential adjuvant in the clinical management of neurological damage caused by viral infections [89,90]. Mice receiving intraperitoneal injections of Lf after lipopolysaccharide (LPS) challenge during pregnancy exhibited longer gestations as a result of a reduction in IL-6 in the plasma. Such studies have used human recombinant Lf, which has greater effects compared to bovine Lf [91,92]. Endometriosis induced by LPS administrations resulted in an increase in NFκB, tumor necrosis factor alpha (TNF-α) and a reduction in IL-1 in Lf-treated animals [93]. When administered orally or intravaginally to pregnant women with risk of preterm delivery, Lf decreased IL-6 in both serum and cervicovaginal fluids, reduced prostaglandin F2a levels and suppressed uterine contractility, with prolonged length of pregnancy observed [94]. Locci et al., 2013 found that, in a group of women at risk of preterm delivery receiving vaginal tablets of Lf (300 mg/day), IL-6 levels were reduced whereas cervical length increased compared to the non-treated women [95], supporting the view that reduced inflammation induced by Lf could prevent preterm delivery. In addition, preclinical evidence suggests that Lf can modulate the consequences of inflammatory challenges to brain development when administered during pregnancy. Lf supplementation from birth and during lactation reduced brain injury in P3 rat pups injected intracerebroventricularly (icv) with LPS [96], reducing ventricular dilatation, hypomyelination and increasing the diameter of the axons in Lf-supplemented animals. Iba1+ microglia cells were reduced after LPS injury and diffusion tensor imaging (DTI) showed microstructural preservation in the striatum 20 days after LPS injection. The application of proton magnetic spectroscopy (^1^H-MRS) showed that LPS injection at P3 induced metabolic alterations related to injury markers, excitatory neurotransmission and anaerobic metabolism, all of which were reduced with lactoferrin dietary supplementation [97]. 

The iron chelation, anti-inflammatory, antioxidant and antiapoptotic activity of Lf has been documented in models of perinatal hypoxia and neonatal hypoxia-ischemia (HI) [97,98,99]. Zakharova and colleagues (2012), using the apo-form of human lactoferrin, observed multi-organ (including brain) protection following deferoxamine-induced hypoxia in mice [98]. Animals were either injected intraperitoneally (ip) or given Lf orally with a dose of 75 mg/kg. Apo-Lf injected animals showed a 40% increased lifespan; the authors attributed the results to the ability of apo-Lf to stabilize hypoxia-inducible factor 1-alpha (HIF-1α) expression. Sokolov and colleagues (2022) administered 10 mg/kg of recombinant human Lf before or after a hypoxic episode during lactation and observed preserved cognitive function in rats’ offspring subjected to prenatal hypoxia [100]. Following HI in three-day-old rats, Lf-supplemented animals showed decreased cortical loss, hypomyelination and preserved diffusivity parameters assessed by MRI at postnatal day 25 [97]. Allied to this, anti-inflammatory (decreased IL-6 and TNFα) and anti-apoptotic (decreased expression of cleaved caspase-3) effects were observed. In a further study, Sanches and colleagues (2021) tested three different doses (0.1, 1 and 10 mg/kg/day) in the same HI model, and observed that acute metabolic disturbance in the cortex induced by HIP3 was reversed by all doses of Lf [99]. Interestingly, MRS evidenced that the Lf neuroprotective effects were dose-dependent. HI pups supplemented with Lf at 0.1 and 1 g/kg had less glutamatergic excitotoxicity and reduced metabolic failure assessed by MRS. The authors suggested that Lf may have had an optimal effect at a dose of 1 g/kg in the model; however, there is still not a consensus on the most effective dose range for neuroprotection and further research is needed. 

The discussion concerning the most appropriate lactoferrin dosage for administration during pregnancy is a matter of intense debate since clinical and preclinical evidence has indicated dose-response effects of supplementation in pathways related to inflammation, cell death and survival or neuroprotection [99,101,102]. Li and colleagues 2015 [103] reported that, following ip injection of LPS in mice, lactoferrin administered at doses of 2.5, 5, and 10 mg/kg attenuated histopathological changes in the uterus, reduced levels of nitric oxide (NO) and inhibited the activation of NF-κB, TNF-α and interleukin-1β (IL-1β) in a dose-dependent manner, with the highest dose showing the best results. Chen and coworkers (2021) reported that normal piglets receiving Lf at a dose of 155 mg/kg/day (suggested high dose) showed better performance in spatial tasks and distinct gene regulation [104]. In the study, lower Lf concentrations modulated genes associated with cellularity and cognition. The authors suggest that potential clinical applications of bovine milk Lf used in infant’s formula should be at low concentrations (~0.5 g/L) to support neurodevelopment, while high concentrations (~1 g/L) should be considered for neuroprotection [102]. Kaufman and colleagues (2020) administered a dose of 100, 200, or 300 mg/kg/body weight of bovine lactoferrin (bLf) to very low birth weight (VLBW) infants enterally for up to 30 days, and reported no adverse effects in preterm babies [101]. In contrast, Dobryk and colleagues (2022), after prophylactic enteral use of bLf for the prevention of severe neonatal diseases in premature infants (gestational age ≤ 32 weeks, birth weight ≤ 1500 g) at a dose of 100 mg/day for at least four weeks, despite reduced hospital length stay of smaller infants, observed no prevention of neonatal diseases (late-onset sepsis, NEC, periventricular leukomalacia and others) in the premature infants receiving Lf and pointed out the urgent need to establish an efficient dose [105]. When Lf was tested in models of IUGR using caloric restriction, oral supplementation with bovine Lf at a dose of 1 mg/kg/day during pregnancy and lactation resulted in cortical microstructure preservation [74]. In a study mimicking pregnant women at risk of premature delivery using rats exposed in utero to exogenous glucocorticoids (dexamethasone), Lf supplementation to the dam through gestation and lactation reversed the effects of injury. Lf prevented glutamatergic dysfunction and restored hippocampal gene expression of brain-derived neurotrophic factor (BDNF) and divalent-metal transporter 1 (DMT-1). Transcripts, such as Nrep (neuronal-regeneration-related protein) and S100B (a glial marker of nervous system damage), were specifically upregulated by maternal Lf in dexamethasone-exposed pups [106,107]. The conflicting evidence available indicates that the determination of efficient (but also safe) dosages, appropriate routes of administration and stage of gestational supplementation requires intense investigation in forthcoming years.

## 4. Early Triggers of Neurodegeneration in Preterm Infants: Protective Roles of Lactoferrin 

Preterm delivery is associated with a high burden of neurodevelopmental impairments and lifelong functional disabilities in distinct cognitive and motor domains [107]. Preterm infants are not adapted to the ex-utero environment and experience reduced access to nutrition, higher amounts of oxygen exposure, blood pressure fluctuations, immune system challenges, and exposure to pro-inflammatory and stressful environment with increased levels of light, sounds, tactile stimulus, olfaction, oxygen and nutrients which are different from those they would have experienced in utero [108]. The developing brain is particularly sensitive to appropriate levels of such factors and preclinical data has shown that alterations in any of these variables can result in abnormal structural and functional development of the brain [109,110]. The CNS has unique characteristics that contribute to the vulnerability of the preterm brain to injury; during this highly active phase, initiation of myelination, axonal and dendritic growth, synaptogenesis and proliferation of microglia and astrocytes are still in progress. Injuries at this stage of development have a profound impact on oligodendrocyte precursors (pre-oligodendrocytes) which are susceptible to injury and cell death, resulting in hypomyelination [111]. The relative vulnerability of this cell lineage is especially mediated by its high susceptibility to oxidative stress and decreased antioxidant defense capacity [78]. In a period in which increased oxygen demand and rate of production of reactive oxygen species is high, oxidative stress and lipid peroxidation are observed in the placenta of women experiencing pre-eclampsia. Newborns also have low levels of endogenous antioxidant capacity, including lower levels of plasma antioxidants, which increase their vulnerability to elevated levels of reactive species. In addition, the preterm brain has comparatively underdeveloped vasculature, which matures relatively late in gestation, thus favoring the occurrence of periventricular hemorrhagic infarction, parenchymal bleeding, with ventricular bleeding and subsequent dilation or hydrocephalus leading to periventricular leukomalacia, poor myelination and cortical development alterations which are associated with significant cognitive and/or motor abnormalities in survivors [112].

Preterm babies are commonly exposed to infections/inflammation, poor respiratory function after birth, ventilation-induced lung and brain injury and a variety of treatments with potential side-effects, including glucocorticoids [113]. The nature of preterm brain injury presents many challenges in the quest to develop treatment strategies for these infants—not only the acute damage should be targeted but supporting subsequent brain development should also be part of therapeutic strategies. It is likely that different treatment strategies may be more effective in treating brain damage stemming from insults such as oxidative stress, ischemia and, particularly, inflammation. Microglial cells undergo critical stages in development throughout the early life period, and, therefore, inflammation can have severe consequences on the proper development of the CNS [114]. Apart from the risk of infections and other specific inflammatory events during pregnancy, delivery and prematurity, other factors may also be related to neurodevelopmental problems, such as obesity, insulin resistance (gestational diabetes) and poor diet; it is well established that insulin resistance is caused by a chronic inflammatory response, and that poor quality diets are major contributors to neurodevelopmental delay in preterm infants [113,115].

A major concern for optimal development and lifelong wellbeing is that of perinatal infections; neonatal sepsis causes 13% of infant deaths and is responsible for 42% of deaths in the first week of life [116]. Even with advances in healthcare reducing the rates of death, infections can still have lifelong consequences; 40% of survivors from neonatal sepsis and meningitis present some form of neurodevelopmental sequelae [117,118]. Necrotizing enterocolitis (NEC) has a mortality rate of 20% to 30% and a quarter of survivors present with microcephaly [117]. Recently, a 17-year study in Denmark and The Netherlands [116] observed that neonates exposed to Group B *Streptococcus* disease (iGBS) had increased risk of neurodevelopmental impairment (RR 1.77 and 2.28 respectively). iGBS poses an even greater burden on developing countries, since not all have similar conditions for prophylactic treatment with antimicrobials. These neonatal pathologies contribute significantly to global inflammation in preterm infants, representing risk factors for brain damage and neurodevelopmental delay. In a complimentary way to human cohorts, animal models enable strengthening of the causal link between perinatal inflammation and neurodevelopmental impairment [119,120,121]. The administration of immunostimulants, such as lipopolysaccharide (LPS) and poly I:C, to pregnant dams has been used to model congenital neurological disorders showing social abnormalities associated with autism spectrum disorder [119,122], cerebral palsy [123,124], white matter injury [78], and a large array of behavioral abnormalities, according to the gestational age and regimen used [125,126]. Postnatal administration of LPS is also often used to model abnormalities related to neurodevelopment [119,122], with outcomes being dependent on postnatal age, dose and regimen. It is important to point out that lipopolysaccharide and poly I:C are molecules targeted by the immune system as pathogen signals, but do not present infectious potential to the organism—LPS is extracted from gram-negative bacterial membranes and poly I:C is a synthetic molecule that resembles viral double-stranded DNA. Both are recognized by toll-like receptors, TLR4 and TLR3, respectively, and elicit innate immune responses [127,128]. Therefore, the effect observed in the offspring is a product of the mother’s or their own immune response rather than microbial damage. This is an important distinction that can help understanding of the implications of the immune system during development, which can be affected by factors other than infections, such as maternal obesity and diabetes, autoimmune diseases, and environmental stresses from drugs, pollutants or malnutrition.

## 5. Lactoferrin and the Development of Infant Microbiome

In recent decades, there has been increasing interest in understanding the importance of commensal bacteria as a major environmental component that promotes healthy development, and is not only related to infectious diseases [127]. The gut microbiota is essential in shaping the development of both the immune and the nervous system [129] through a symbiotic interaction with the host [130], providing environmental “clues” that influence immune system development, including pathogen-associated molecular patterns (PAMPs), such as LPS, MDP-2 (muramyl dipeptide 2) and LPA (lysophosphatidic acid), that activate mucosal TLRs, which drive immune system development [131,132,133]. Indeed, it has been proposed that newborns present a reduced immunological response compared to adults because their gut may be sterile during gestation, and therefore “naïve” to pathogens [134]. Animal models of germ-free mice (GF), which are delivered under fully aseptic conditions, are more susceptible to infections, having a reduced number of Th17 cells which are particularly important for immunomodulatory function in the lamina propria. Moreover, GF mice present abnormal behavior [135,136], immature microglia [130,137,138] and abnormal intestine sizes. Since microglia are responsible for synaptic pruning during brain maturation, such hypofunction has been linked to future psychiatric diseases [131,132,133]. The gut microbiome also provides the host with metabolites which would otherwise be unavailable, such as short-chain fatty acids (SCFA) (e.g., butyrate, propionate and acetate) [139]. These molecules are an exclusive product of fiber digestion by these bacteria since humans lack cellulase enzymes. These molecules can be further oxidized for energy production, but also have signaling properties that modulate immune system development [140], especially because these molecules can easily cross the intestinal barriers and diffuse across the body, regulating inflammation and immune responses. Their impact is highlighted by the number of recent studies investigating SCFA administration to treat diseases [140], and how this links fiber-rich diets and human health [141,142].

A newborn’s own microbiome development is started from their own mother’s; vaginal delivery is a major event in this process of microbial transfer. Babies delivered by cesarean section miss this step and may be more prone to have allergies, asthma and slower immune development [143,144]. Interestingly, it has been observed that the vaginal microbiome of preterm mothers differs from that of term mothers, with a reduced abundance of beneficial microbes, such as *Lactobacillus* [140]; it is possible that such imbalances contribute to proinflammatory cytokines and could increase preterm delivery probability [140]. It has been speculated that breastfeeding is another important process shaping the newborn microbiome since skin-to-mouth contact during suckling allows the entry of beneficial microbes derived from the mother. Furthermore, the composition of colostrum and milk have a determinant role, since diet is the main driver in microbiome formation and maternal milk likely has a unique balance between lipids, proteins and oligosaccharides that contribute to microbiome development [145] (Figure 2). In fact, evidence suggests that formula-fed infants are more susceptible to allergies and have lower microbiome diversity. This is probably because formulas up to this time are more focused on macronutrient composition but do not fulfill the possible “functional” roles of maternal milk [125]. In this context, lactoferrin has been shown to have many roles in regulating immune function, whether endogenously produced or from the diet. 

It is well known that nutrition is the main determinant in the establishment and maintenance of the gut microbiota [146]. Gut maturation is directly associated with commensal microbes. Mucus, which provides a protective barrier against pathogens, is also dependent on the microbiome, which provides the building blocks for mucus production. Mucus also participates in intestinal transit, allowing better absorption of nutrients that are important not only to the organism but to the enterocytes themselves. The maternal vaginal microbiota has been associated with recurrent and increased risk of premature delivery [140,147], and the newborn microbiome seems to develop differently according to gestational maturity at birth [148] predicting the risk of brain damage in preterm infants [146,149]. However, the exact mechanisms connecting the microbiome to brain health are largely unknown. In the following postnatal weeks, the microbial population develops in diversity as well as abundance and is mostly determined by nutrition. Breastfeeding has been associated with greater bacterial diversity in the gut and protection from allergies, colic and autoimmune conditions [146,150]. Interestingly, one of the major factors associated with NEC is the use of enteral formula instead of human milk [151]. It is possible that, besides the mother’s own antibodies transferred to the infant, some macromolecules present in milk are not meant for the infant, but are to develop certain bacterial genera that can produce their own beneficial metabolites for the host, such as SCFA, amino acids and vitamins [146,149]. SCFA, such as butyrate and propionate, are the most studied bacterial metabolites, and have been demonstrated to be neuroprotective in neonatal ischemia [152,153] and to promote microglial maturation [130,154]. They are mostly produced by bacteria from oligosaccharides that are not digestible by human gut enzymes [35].

Lactoferrin, as well as being present in milk and secretions, is also found in the granules of polymorphonuclear neutrophils [155] which are recruited to the small intestine during gut inflammation and can release Lf as well as other iron-sequestering proteins, such as NGAL (lipocalin 2) [156,157,158]. Iron sequestration from the milieu is a conserved strategy for fighting bacterial infections since most bacteria depend on Fe3+ as a cofactor for energy production and DNA replication [156]. Therefore, iron depletion from the media hampers bacterial survival. Lactoferrin is specially designed for this function since its affinity for iron can be 300 times higher than that of serum transferrin [159], and its affinity for iron is maintained even at low pH, such as in the small intestine [159,160]. Lf iron-scavenging activity is not its only role against bacterial infections, especially since some bacteria have evolved mechanisms to use other metals, such as zinc, instead of iron, or to secrete iron siderophores that have higher affinity than those of transferrins for bacterial uptake. It has been demonstrated that Lf can disrupt the formation of bacterial biofilms by *Pseudomonas aeruginosa*, which is a key mechanism by which the bacteria resist antibiotic treatment [158,161,162]. Another major role of Lf as an antimicrobial agent is its ability to bind to LPS [92,96], which has two important implications: it can disrupt bacterial membranes and also reduce LPS binding to the toll-like receptor 4 (TLR4) [96,163]. This latter property is important since it has been demonstrated that TLR4 is a major component in NEC-associated neurological sequelae [164]. Therefore, it is possible that Lf can also prevent exacerbated inflammatory responses to bacterial infections. 

In addition to its role as an antimicrobial agent, it has been demonstrated that Lf also has a role in immune function and has anti-inflammatory properties [76,88]. In mouse models it has been observed that oral administration of Lf increased cell counts of CD4 CD8 T-cells and NK cells in the lamina propria of the small intestine [165,166]. In addition, lactoferrin receptor expression has been observed in several subsets of immune cells, although its role is still unclear [167,168] due to the capacity of Lf to bind to a multitude of molecules. In vitro studies have demonstrated that Lf likely reduces the release of pro-inflammatory cytokines [169]; however, it is necessary to investigate further what causes such effects, since Lf may be binding to immunostimulants, such as LPS and Poly I:C—in this scenario, such effects would only apply to that specific model and could not be extrapolated to other situations. When observing in vivo responses to lactoferrin, it is often difficult to differentiate its direct effect on immune response and its indirect effect through antimicrobial properties. Even if it is accepted that bacteria/viruses are not really involved in neuropathology in preterm infants as a primary cause of damage, many inflammatory complications due to hypoxic-ischemic events, and, in the most severe cases, neonatal sepsis and NEC, can present concomitant infections; thus, naturally, the effect of Lf on pathogenic bacteria will also modulate the inflammatory response. Another important consideration is the ability of Lf to bind to the Lipid A fraction of LPS, thus impairing its ability to bind to the CD14/TLR4 complex which promotes microbial pattern recognition and the inflammatory response [93,170]. Although TLR4 is an important element in the immune response for the detection of bacterial pathogens, its persistent stimulation by endotoxins can promote tissue damage through an exacerbated immune response. This has been demonstrated in NEC, in which TLR4 stimulation has been linked to both intestinal and cerebral damage in affected preterm infants [164]. TLR4 activation by commensal microbes is a necessary process in the development of the immune system but is also involved in the pathology of infections and it is likely that Lf present in milk is particularly adapted to allowing LPS-TLR4 binding in the physiological range.

It was discussed previously how the gut microbiome can be important for neurodevelopment, and how dietary lactoferrin present in a mother’s milk could be helpful in shaping a beneficial microbiota, which in turn can prevent neurodegeneration; however, such a relationship still needs further study. AD patients present alterations in the gut microbiome when compared to controlled pairs; however it is important to point out that there is far from a consensus on what constitutes a “normal” microbiome. Case studies of fecal microbiota transplantation to patients with *Clostridium difficile* gut infection that also show AD suggest that microbiome transplantation of a “healthy” form could improve cognitive symptoms [171,172]. Moreover, mouse models of AD, when colonized with the microbiome from their wild type controls, show alleviation of disease progression [173]. Gastrointestinal symptoms, such as constipation, occur in Parkinson’s disease patients and probiotics are prescribed to alleviate those symptoms [174]. It has also been suggested that the gut microbiome could be involved in the progression of the disease [175,176]. If so, lactoferrin could represent a dietary approach for treatment in these patients, by playing an active role in the gut, regulating iron homeostasis and acting as a source of dietary protein.

## 6. Lactoferrin for Preventing Neurodegeneration: A Promising Molecule?

Although most Lf research is focused on the neonatal period due to the presence of Lf in mothers’ milk, new evidence has shown benefits of Lf in neurodegenerative diseases (Figure 3). Lf is present in the CNS and can easily cross the blood-brain barrier from the peripheral circulation [177,178] since some of its putative receptors are found in the brain endothelium, such as LRP1 [179,180] (LDL receptor-related protein 1) and intelectin-1[160], facilitating transcytosis into the brain parenchyma. For this reason, Lf structural motifs are being studied for application in liposomes for targeted CNS delivery of drugs, and these receptors could be used for mediating Lf-bound drugs entry into the CNS [181,182].

Concerning its role in neurodegeneration, Lf seems to be increased around amyloid plaques in post-mortem Alzheimer’s disease (AD) brains [171] and in APP-transgenic mice (specially in animals older than 20 months), which resemble the later stages of AD [172]. The purpose of this accumulation around amyloid deposits remains unknown; recently, the APOE ε4 allele [183], which is strongly related to the onset of AD, has been related to abnormalities in brain iron homeostasis and the induction of ferroptosis [183,184]. In this paper, although lactoferrin was not mentioned, other transferrins were studied. It is possible that early data regarding Lf around plaques could reflect the process of iron dysregulation, but this mechanism is still unknown. Recently, the neuroprotective effects of intranasal administration of lactoferrin have been tested in an APPswe/PS1DE9 transgenic mouse model of AD [185]. Intranasal human Lf (hLf) reduced β-amyloid (Aβ) deposition and ameliorated cognitive decline in AD animals through ERK1/2-CREB and HIF-1α signaling pathways activation and consequent ADAM10 expression. The intranasal route is a relatively new and very interesting approach due to increased and target-specific availability in the CNS [186]. 

Recently, Lf has been emerging as a putative biomarker for the early diagnosis of neurodegenerative diseases. Salivary Lf was reduced in patients with AD and MCI, as confirmed by PET imaging of amyloid using ([^18^F]-florbentapir) [187], and one study has suggested a negative association with amyloid load [188,189,190]. However, there is far from a consensus on this, since another study failed to observe such a relationship [191], due, at least partially, to more heterogeneous samples and the diagnostic criteria used in the different studies. The causal relationship between salivary lactoferrin (sLf) and AD is still unknown; however, it has been hypothesized that reduced sLf may reflect immune system abnormalities, which are commonly observed in AD patients [188,192]. A major caveat in these correlations is that immunological disturbances are observed across most neurodegenerative diseases [192] and even chronic diseases that are not directly associated with the CNS show immunological alterations during their onset [193]. In this sense, Gonzales–Sanchez data suggests that reduced sLf is specific to AD pathology; therefore, there must be a specific mechanism linking peripheral levels of Lf and AD physiopathology [194]. 

In Parkinson’s disease models, Lf is increased in rat brains after MPTP administration [195], which was suggested to be a protective tissue response, since iron dysregulation is observed in PD [195,196]. Iron can cause oxidative stress through the Fenton reaction, and it is possible that one of the Lf neuroprotective mechanisms is related to iron chelation. Treatment with oral hLf has been demonstrated to be neuroprotective in an MPTP rodent model of PD, as well as in MPP+ in vivo and in vitro models [195,197,198]. Interestingly, these studies used both apo and holo forms of Lf and observed no difference between them. Thus, it is reasonable to assume that some benefits of Lf are independent of its iron scavenging properties [186]. Data is still scarce in humans, with a limited number of studies using post-mortem tissue. Alzheimer’s [199] and Parkinson’s disease mice models presented abnormal microbiomes when compared to wild type and improved their symptoms following antibiotic-induced elimination of the gut microbiome [175,200]; this new evidence suggests lactoferrin is a promising candidate for preventing the evolution of neurodegenerative diseases. 

## 7. Conclusions

CNS development is a long and complex process in which the best strategy for protecting neonatal brains (especially preterm ones) has yet to be established. Due to their role in “shaping” the developing brain, nutritional interventions can be helpful in clinical practice and, since it is unlikely a single molecule will be able to protect the entire neurodevelopmental trajectory, interventions could include different strategies to prevent risk factors. Lactoferrin is suitable for these purposes either by preventing preterm delivery or by treating preterm birth-related disabilities, particularly by decreasing inflammatory processes during pregnancy. Lf could also optimize brain development during a period of high cerebral vulnerability by preventing abrupt alterations in the preterm neonate’s environment, which makes it a cost-effective approach in the long-term. A better understanding of the role of the different forms of lactoferrin (i.e., apo or holo) must be considered and should receive as much attention as the dose and route to be administered, and the timing and duration of treatment during pregnancy, to maximize the effects of preventing not only preterm delivery but also neurodevelopmental long-term outcomes. When studying lactoferrin as a treatment option for the prevention of neurodegenerative diseases, researchers should address fundamental questions, such as its distribution across the CNS, its bioavailability and its molecular targets (which are currently not totally known). It is important also to highlight that studies assessing salivary Lf as a biomarker could help elucidate the role of lactoferrin in Alzheimer’s disease and other dementias, especially when studied in combination with established biomarkers of neurodegeneration.

## Figures and Tables

**Figure 1 nutrients-14-02923-f001:**
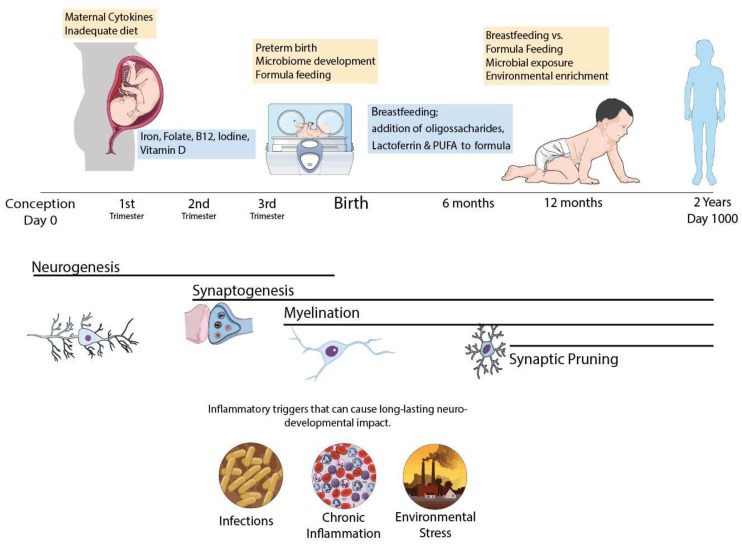
Schematic representation of the CNS development in the first 1000 days period. Infant’s brain undergoes major hallmarks of neurodevelopment during this period and can be affected by poor nutrition, infectious pathogens and pollutants with life lasting consequences. Common neurodevelopmental disorders involve myelination disturbances (IUGR, prematurity) or synaptic pruning failure such as Autism Spectrum Disorder and Schizophrenia and can be triggered by early life events during brain maturation. Nutritional policies can be adopted to improve outcomes, with proper macro and micronutrient adequacy. Specific macronutrients such as Lactoferrin and n-3 polyunsaturated fatty acids have promising benefits for neurodevelopment and should be further explored in clinical trials.

**Figure 2 nutrients-14-02923-f002:**
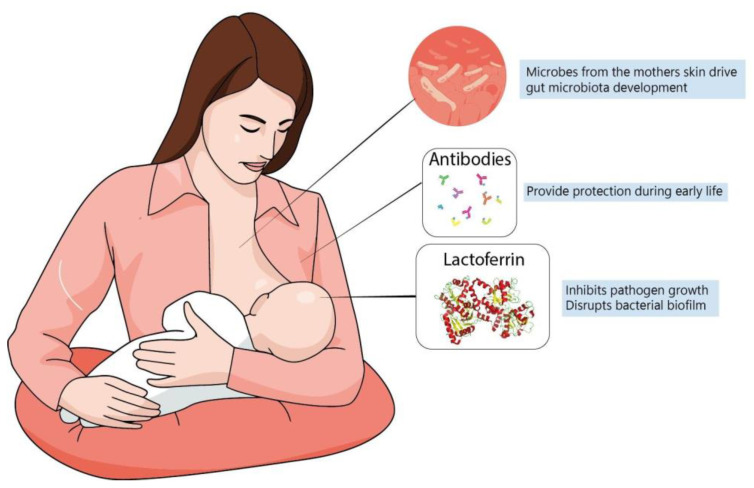
Maternal milk is the optimal source of nutrients for newborns, even in the absence of consensus about its ideal composition. Dietary interventions during pregnancy can alter maternal milk composition and alter the infant’s immune system, playing a major role during brain development. The best understanding of molecules present in maternal milk is important not only for improving recommendation directives, but also for defining better formulas to be administered for infants at risk.

**Figure 3 nutrients-14-02923-f003:**
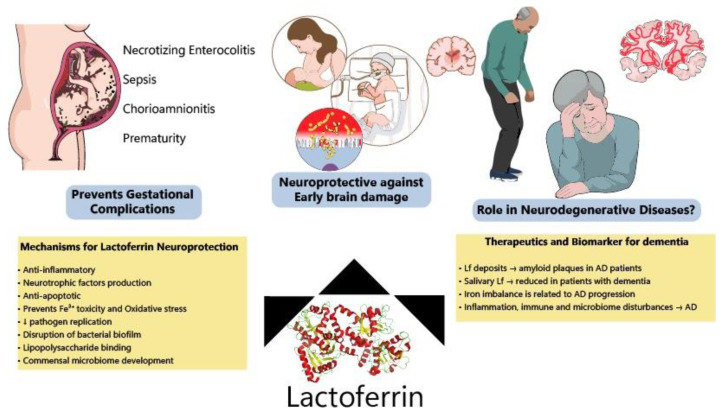
Protective effects of lactoferrin through the lifespan. Lf has been shown to prevent disturbances during pregnancy, decreasing the incidence of NEC, sepsis and prematurity among other complications. Allied to this, clinical and preclinical data have indicated that Lf modulates several pathways (ion scavenger, anti-inflammatory, anti-apoptotic and improving microbiome) protecting the developing brain. Lf has also been considered a potential biomarker of neurodegeneration and its mechanisms of action make it a good candidate to be tested for the prevention of neurodegenerative diseases.

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
