# Peer review of "Neuroprotective Role of Lactoferrin during Early Brain Development and Injury through Lifespan"

_nutrients, 2022, doi:10.3390/nu14142923_

Round 1

Reviewer 1 Report

This review highlights how nutrients benefit early stages of neurodevelopment. It points toward the beneficial role of Lactoferrin, an iron binding protein present in colostrum and human milk, during breastfeeding in postnatal development and cognition. Anti-inflammatory and antimicrobial properties are the most-studied of Lactoferrin reducing sepsis and necrotizing enterocolitis in premature births. Lactoferrin is able to exert neuroprotection in the developing brain, enhance brain connectivity, neurotrophins production and decrease inflammation in animal models of perinatal inflammation, intrauterine growth restriction and neonatal hypoxia-ischemia. Authors address the relationship of lactoferrin and the development of infant microbiome. Authors also discuss the scarce evidence linking Lactoferrin with the prevention of neurodegeneration.
Although the review is very important contribution to the field, some points need to be addressed:

Explain in more detail the differences between apo and holo forms of Lactoferrin.

Scientifics names of organisms such as bacteria, should be written in Italics

Lactoferrin and other words were not abbreviated throughout the manuscript.

Figures are not inserted in the text.  

Lines 207-210. Revise the meaning of these sentences.

Lines 435-439. The idea is repeated in lines 407-415

Lines 464-466. The idea is repeated in lines 455-457

Author Response

Authors thank to the reviewer for the contributions.

The sentence "  Due to its iron-binding properties, Lactoferrin can exist in the so called Apo (iron-free) and holo when saturated with iron, with a mixture of both being observed in milk [83,84]. This distinction is important because apo-Lf can easily chelate iron preventing bacterial growth, whereas holo-Lf is superior correcting iron deficiency [83]." regarding holo and Apo forms of Lf were added.

Scientific names were changed to the latin form.

Figures were inserted in the format sent by the journal.

The manuscript was entirely reviewed for an English Native speaker and the meaning of the sentence highlighted by the reviewer as well as repetitions observed, were adjusted. 

Reviewer 2 Report

Thank you for the opportunity to review this manuscript. The topic is very interesting.

The paper is well-written, but the fact that is a narrative review should be specified.

I suggest some other improvements.

News of the probable usefulness of Lactoferrin in the treatment of different diseases and in the prevention and treatment of Covid-19 was reported. The reasons why Lattoferrin was investigated in this area are essentially 2:Milk (maternal and bovine) and colostrum are foods particularly rich in Lactoferrin and it has been observed clinically that infants have suffered less heavily from SARS-Cov-2 infection.

Lactoferrin had already been studied on several viruses, including the virus responsible for the 2002 SARS epidemic, genetically similar to the present virus. However in this manuscript, although all possible applications are described in the relationship of the neuroprotective role of Lactoferrin, the description of the important antiviral activity of lactoferrin is missing. The recent emergence of the Coronavirus disease-2019 (COVID-19) pandemic, initially considered a respiratory illness, demonstrated a broader virulence spectrum with the ability to cross the blood-brain barrier and inflict a plethora of neuropathological manifestations in the brain - the Neuro-COVID-19.  I think it's very important to introduce these concepts. Lactoferrin, credited with several neuroprotective benefits in the brain could serve as a potential adjuvant in the clinical management of viral infection. Good references are doi: 10.1002/rmv.2261 and doi: 10.1016/j.jiac.2014.08.003.

Author Response

Authors thank to the reviewer.

The description of the manuscript as a narrative review was included in the abstract.

Authors included the following sentence regarding the effects of Lf and viral infections in the manuscript: " Additionally, Lf has been tested for its anti-viral properties, improving outcomes of infection models of HSV, Influenza, Rotavirus and RSV. Lf seems to inhibit viral replication in vitro for HSV-1 and seems to reduce infection by the SARS-CoV 1 & 2, both by interacting with cell surface proteoglycans and could serve as a potential adjuvant in the clinical management of neurological damage caused by viral infections [89,90].  "

Comments inserted are highlighted in blue and sentences modified, highlighted in yellow.